# Red Marine Algae *Lithothamnion calcareum* Supports Dental Enamel Mineralization

**DOI:** 10.3390/md21020109

**Published:** 2023-02-02

**Authors:** Marcela R. Carrilho, Walter Bretz

**Affiliations:** 1College of Dental Medicine-IL, Midwestern University, Downers Grove, IL 60607, USA; 2TeraEarth, Huntington Beach, CA 90742, USA

**Keywords:** marine natural products, calcareous seaweed, bioactivity metabolites, *Lithothamnion calcareum*, Brazilian coastline, dental erosion

## Abstract

The current management of oral conditions such as dental caries and erosion mostly relies on fluoride-based formulations. Herein, we proposed the use of the remaining skeleton of *Lithothamnion calcareum* (LC) as an alternative to fluorides. LC is a red macroalgae of the Corallinales order, occurring in the northeast coast of Brazil, whose unique feature is the abundant presence of calcium carbonates in its cell walls. Two experimental approaches tested the general hypothesis that LC could mediate enamel de-remineralization dynamics as efficiently as fluorides. Firstly, the effect of LC on enamel de-mineralization was determined in vitro by microhardness and gravimetric measurements to test the hypothesis that LC could either prevent calcium/phosphate release from intact enamel or facilitate calcium/phosphate reprecipitation on an artificially demineralized enamel surface. Subsequently, an in situ/ex vivo co-twin control study measured the effect of LC on the remineralization of chemical-demineralized enamel using microhardness and quantitative light-induced fluorescence. With this second experiment, we wanted to test whether outcomes obtained in experiment 1 would be confirmed by an in situ/ex vivo co-twin control model. Both experiments showed that LC exhibited equivalent or superior ability to modulate enamel de-remineralization when compared to fluoride solution. LC should be explored as an alternative to manage oral conditions involving the enamel demineralization.

## 1. Introduction

Dental enamel is a tissue-assembled and specialized natural biomaterial that plays unique structural, mechanical, and esthetic roles in the complexity and variety of functions of the dental organ. Given the global burden of oral disorders such as dental caries and erosion [1,2], oral care providers are often intervening to prevent, reverse, or treat the excessive non-physiological loss of minerals from the enamel. Handling enamel and issues related to its homeostasis are, therefore, part of the routine of several dentists and oral hygienists around the globe.

Despite representing completely different entities, with specificities that go beyond the etiology, phenotype, and mechanism of progression, dental caries and dental erosion share a common event which is a time-dependent demineralization/dissolution of tooth hard structures (i.e., enamel, dentin, and cementum). Due to anatomical reasons, enamel is in the frontline of dental demineralization; from a homeostasis standpoint, the partial dissolution of enamel is key to reinstate the re-equilibrium of mineral content (i.e., calcium phosphates) in oral fluids [3,4,5].

Over the past 80 years, fluoride-based formulations have been systematically employed to regulate the mineral imbalance between mineralized dental hard tissues and surroundings, hence serving to delay the progression of dental caries lesions [6,7]. The topical application of fluorides has also been used to manage dental erosive lesions [8,9]. Given the fact that fluorides have a long history in dental demineralization management, the literature is populated with a number of studies on the mechanism, outcomes, advantages, pitfalls, and controversies related to the application of fluorides in dentistry [10,11,12,13,14]. 

While the side effects of fluorides on a systemic level are yet to be proven [15,16], an effervescent worldwide debate challenges their use as the major epidemiological approach in combatting dental caries [17,18]. In recent years, many of the member states of the European Union and a few municipalities in Canada and in the US have enforced strictive regulatory measurements or promoted public opinion consultations concerning the intake of fluorides through community water fluoridation [19,20,21]. The profusion of beliefs in favor and against the deliberate and continuous administration of fluorides evidences a rhetorical dispute that is far from finding a final resolution. At the same time, such a controversial contest also offers a unique opportunity to seek therapeutical alternatives that can provide safe and effective means to address the demineralization-related dental disorders. 

In the present study, the biotechnological potential of the remaining skeleton of the macroalgae *Lithothamnion calcareum* (LC) was tested in its ability to modulate the enamel de-remineralizing dynamics. LC is a red macroalgae of the Corallinaceae family, which is abundantly found on the Brazilian coastline. A unique feature of LC is the presence of calcium and magnesium carbonate precipitates in its cell walls. Additionally, the LC remaining skeleton consists of more than 20 oligoelements and bioactive molecules (i.e., polyphenolic compounds and terpenes) with relevant therapeutic properties [22,23,24,25,26,27]. The rationale behind the selection of LC to control and manage enamel mineralization mainly lies in its high calcium content and documented use as a nutraceutical supplement to mitigate bone destruction and support bone repair and turnover [23,24,25,26,27,28].

Based on the principle that enamel remineralization can be enhanced when it gets exposed to saturated calcium delivering sources [29,30,31], we hypothesized that such calcium-rich marine extract could either prevent calcium/phosphate release from enamel or facilitate calcium/phosphate reprecipitation on the enamel surface.

## 2. Results

Two experimental approaches tested the general hypothesis that LC could mediate enamel de-remineralization dynamics as efficiently as fluorides. Experiment 1 consisted of an in vitro study that tested the effect of LC on enamel de-mineralization of intact and demineralized enamel samples by Knoop microhardness (KHN) and gravimetric measurements. In experiment 2, an in situ/ex vivo co-twin control study measured the effect of LC on the remineralization of chemical-demineralized enamel by KHN and quantitative light-induced fluorescence (QLF).

### 2.1. Experiment 1—In Vitro Study

The two panels in Figure 1 show the outcomes relative to intact (upper panel—Figure 1A–D) and caries-challenged (pre)demineralized (lower panel—Figure 1E–H) enamel samples upon treatment with the remaining skeleton of LC in two concentrations, in comparison to positive (0.05% sodium fluoride—NaF) and negative (water—W) controls. The effect of enamel treatment was tested on the variables: surface microhardness change (%SHC) of intact enamel, surface microhardness recovery (%SHR) of (pre)demineralized enamel, and cross-sectional hardness and mass variation (%Mass change or %Mass Recovery) of both intact and (pre)demineralized enamel samples.

Both of the tested LC solutions (0.01 and 0.03%—01 LC and 03 LC) attenuated the demineralization of intact enamel (Figure 1A–C), and promoted an increase in the surface and cross-sectional hardness of demineralized enamel (Figure 1E–G). The mass of intact enamel samples treated with LC or NaF increased noticeably in comparison to the negative control sample (i.e., specimens treated with W) (*p* < 0.05) (Figure 1D). No statistical difference was detected in % mass change after treatment among intact dentin samples treated with 0.03 LC and NaF (*p* > 0.05) (Figure 1D). Likewise, mass recovery of demineralized specimens was only achieved with LC and NaF treatments (*p* > 0.05) (Figure 1H). Overall, regardless of the tested outcome, no statistical differences were found between the effects of 0.03% LC and 0.05% NaF (*p* > 0.05) for both intact and demineralized enamel. 

### 2.2. Experiment 2—In Situ/Ex Vivo Co-Twin Control Study

There were no adverse reactions reported by participants while undergoing study protocols. Similarly, there were no effects of study protocols on oral tissues. Surface hardness recovery (%SHR) and quantitative induced-fluorescence (QLF) measurements are summarized in Figure 2. %SHR of demineralized samples treated with LC 0.03% was significantly higher when compared to positive control NaF 0.05% (*p* < 0.05) (Figure 2A). Likewise, LC 0.03% solution was significantly more effective in increasing surface hardness of chemically demineralized enamel than placebo solution (*p* < 0.01) (Figure 2B). QLF parameter delta F (∆F) refers to the amount of fluorescence signal which allows for quantitation of mineral loss, while delta Q (∆Q) is derived from the product of ∆F (in %) multiplied by the lesion area. The treatment of demineralized enamel for the first set of twins showed no statistical difference between LC and NaF groups for ∆F and ∆Q values (Figure 2C,D) (*p* > 0.05); although, there was a tendency for improvement in ∆F and ∆Q values in the LC group when compared to NaF. Conversely, a remarkable difference was observed for the second set of twins who rinsed with LC or W (placebo), where it was seen that ∆F and ∆Q (Figure 2C,D) values for the LC group were significantly lower than for the group who rinsed with water (*p* < 0.05). 

## 3. Discussion

Collectively, present experiments showed that 0.03% LC reduced mineral loss and promoted mineral gain, respectively, in intact and (pre)demineralized enamel in both of the present experiments, in vitro and in situ/ex vivo. Accordingly, the general study hypothesis that the remaining skeleton of calcium-rich marine macroalgae LC could both prevent calcium/phosphate release from enamel and favor calcium/phosphate reprecipitation on the enamel surface cannot be rejected. 

The process of demineralization and remineralization of dental hard tissues is finely governed by the degree of saturation of oral fluids with apatite minerals [29,30]. In physiological conditions, dental enamel demineralization is reversed if sufficient time passes between acidogenic challenges, hence allowing remineralization. This is what in vitro pH-cycling models aim to replicate. They simulate not only pH fluctuations over specific periods of time (i.e., with periods of demineralization interspersed with remineralization), but also ensure that dental enamel is bathed in fluid phase saturated with optimal levels of calcium, phosphate, fluoride, and other ions composing the lattice of apatite minerals to enable remineralization and maintenance of enamel surface integrity [32,33]. Experiment 1, based only on laboratory models, was designed to provide a preliminary screening on the potential for LC to be used to manage the process of enamel de-remineralization.

The anti-demineralizing effect of calcium/phosphate-based solutions and biomaterials is thought to occur when chemical driving forces in oral fluid milieu boost up apatite precipitation on tooth structures. Minerals found either in saliva, delivered from mouth-rinses and mineral-based biomaterials, or supplied via food/liquid ingestion can be dissolved in salivary acids [29] and be made available for dental substrate remineralization. More specifically, carbonic acid in oral fluids quickly converts itself to carbon dioxide and water [34], which is key for mineral availability. When these events take place, the dissolved mineral ions become prone to reprecipitate as solid mineral ions on the dental surface, hence enhancing the remineralization stage [3,4,5]. Since this process does not imply any biological activity, it is commonly described as being a passive mineralization [4,14]. The results of the present study confirmed such premises (Figure 1). The increase in microhardness (surface and cross-sectional) and mineral gain of demineralized enamel upon treatment with LC were encouraging and set the ground for us to venture out and perform an in situ/ex vivo experiment. 

By using a co-twin control model in experiment 2, we basically confirmed that 0.03% LC could effectively promote enamel remineralization. The increment of surface hardness in demineralized enamel specimens treated with 0.03% LC solution was about three-fold greater in specimens treated with 0.05% NaF. Conversely, the placebo mouthwash (sterilized water—W) did not promote significant hardness recovery during the two-week intervention (Figure 2A,B). Likewise, variations in QLF parameters, ∆F and ∆Q, were similar between specimens treated with LC and NaF, and significantly more evident for LC compared to placebo treatment (Figure 2C,D). In addition, QLF results indicated that LC led to an enamel mineral gain that was slightly greater than that obtained with NaF. 

Despite the experiment having not been designed to enable a regression analysis between hardness and QLF data, there is a manifest correspondence between these dependent variables. Similar to hardness measurements, QLF parameters (i.e., ∆Q and ∆F) clearly showed that LC, differently from placebo, was capable to stimulate enamel remineralization. The literature does not express a consensus regarding the accuracy, sensitivity, and specificity of the QLF method for caries lesion detection when compared with other available protocols (i.e., ICDAS, laser fluorescence, transillumination, light-emitting diode devices, fluorescence imaging with reflectance enhancement, digital radiography). While some studies fairly indicated that QLF may exhibit good predicting performance [35,36,37], others did not confirm such a trend [38,39,40]. Nevertheless, since the QLF measures and correlates the difference in fluorescence between a given lesion and surrounding intact enamel in relation with the amount of mineral loss, it has been advocated as being useful to estimate changes in mineral content and lesion size [41]. 

To further ascertain the effectiveness of formulations on enamel mineralization, we designed this co-twin control study. The advantages of a co-twin control study over a conventional, case-control study of unrelated individuals are multi-faceted and include the following: (1) control of environmental factors, e.g. diet, (a central concern in studies of dental caries), lifestyle, control for twin pairs living in the same household; (2) reduction in bias often associated with the use of unrelated individuals in case-control studies; (3) increase in study power since many factors leading to dental caries are controlled for; and (4) increase in cost-efficiency since equivalent conclusion power may be obtained from significantly smaller cohorts [42]. To the best of our knowledge, there is not much use of co-twin control models for the assessment of conditions related to dental mineral deprivation. The present study may represent the first co-twin control approach that took into consideration the multitude of dietary phenotypes (i.e., energy and nutrient intakes, dietary patterns, and specific food group intakes) in the context of enamel remineralization.

All participants that provided us (i.e., 93%) with a diary containing a description on food and drink habits reported to consume from two to four units of sugar-rich snacks/drinks throughout the day on daily basis. We believe that such a high daily consumption of sugar may have impacted this study’s outcomes. However, since most participants described a similar pattern and/or frequency of snack/soft drink consumption, we trust that the individual diet habits of participants would not have likely had a confounding effect on the present results. It is worthwhile mentioning that participants were advised to perform their routine oral hygiene practices after the meals and before reinserting the appliance into the mouth. Moreover, participants’ hygiene practices were shown to be adequate even prior to the study commencement and, likewise, should not be a confounder for the present outcomes. 

We offer a justification for LC to perform similarly (experiment 1) or even better (experiment 2) than fluoride in supporting enamel hardness recovery and mineral gain. The remaining skeleton of seaweed LC is a natural source of calcium (Ca), magnesium (Mg) and several other mineral elements. Even if Mg concentrations in mature enamel are low, and reported to range from 0.4% in the inner enamel layer to 0.1% near the outer enamel surface, its presence in the fluid surrounding enamel is key for enamel development and mineralization [43,44]. In fact, Mg has been shown to exert a marked effect on the structure, size, and mechanical properties of enamel apatite nanocrystals [45,46,47,48]. Accordingly, Mg is expected to play an adjuvant but significant role in therapeutical schemes aiming at either the remineralization of mature demineralized enamel, or prevention of intact enamel demineralization. We speculate that the multi-oligoelemental composition of LC may further act in synergism with Ca and Mg to assist enamel remineralization. In effect, recent epidemiological studies have begun to appreciate the role of trace elements other than Ca in the mitigation of chronic diseases [49,50]. To our understanding, current methods to prevent and/or manage dental enamel demineralization do not even consider such a premise. 

The positive results obtained with the use of LC for the management of bone disorders in animal models [23,24,25,27] are even more motivating. Dietary supplementation with LC of female mice under a high-fat diet was significantly shown to prevent Ca loss (i.e., skeletal demineralization), reduce bone loss, and increase bone stiffness [23,24,25]. Furthermore, a double-blind crossover pilot trial indicated that LC ingestion modulated serum markers for Ca metabolism in premenopausal women, and consequently, could represent an effective means of providing Ca supplementation to individuals at risk of bone loss due to osteoporosis [28]. Besides the abovementioned effects of Mg and other trace elements on bone apatite crystallographic structure and mechanical properties, these studies sustained that dietary LC supplements might involve systemic reactions, such as inflammation control and/or immunological response, which in turn might positively impact the mineral balance/shift between mineralized tissues and their respective bathing fluids. This is a hypothesis that merits further investigation.

Although the decrease in caries experience associated with fluorides was rapid in the early 1970s, it leveled off and essentially reached a plateau in the 1990s [51]. The World Health Organization (WHO) recently reported that about 60–90% of children and over 90% of adults have experienced dental caries [52]. Together, these findings highlight the need for additional and more effective measures to regulate enamel mineral losses caused by dental caries and erosion. While recognizing the pivotal role played in this arena by fluorides over the last eight decades, we trust that it is crucial to look forward and search for natural resources with additional biological cues. The present results provide compelling evidence that a multi-mineral natural approach can be beneficial to target dental hard tissue disorders. We are positive that LC is not only a safe and bioactive agent for the management of dental caries and erosion, but can be also envisioned as a more inclusive oral healthcare promoter.

## 4. Materials and Methods

Reagents employed in this study were obtained from Sigma-Aldrich (Saint Louis, MO, USA) unless otherwise specified. The marine-derived poly-mineral extract is commercially available in Brazil as a dietary supplement (Vitalidade 50+®, Phoster Algamar, RJ, Brazil) and was obtained from the skeletal remains of red marine algae *Lithothamnion calcareum* (LC). The algae thrive in the Atlantic waters off the northeast coast of Brazil. The major elements of LC extract are Ca and Mg (more than 20% and 4% by weight, respectively), but it also contains measurable levels of 74 other trace elements, including phosphorus (Appendix A).

### 4.1. LC Solubilization and Ion Content Characterization

A series of preliminary studies were conducted to determine the solubility of LC in aqueous solution (Appendix A). Two concentrations of LC at 0.01 and 0.03% (*w*/*v*) were selected (pH 6.8) for subsequent testing. The concentrations of Ca, P, Mg, and F in these solutions were evaluated by inductively coupled plasma–atomic emission spectrometry (Appendix A).

### 4.2. Experiment 1—In Vitro Study

#### 4.2.1. Enamel Blocks Preparation and Baseline Measurements

Two hundred and forty (*n* = 240) bovine incisors with complete root formation were collected and immediately stored in 0.9% NaCl solution, containing 0.02% sodium azide at 4 °C for one week. Enamel blocks (4 × 4 × 2.5 mm thick) were obtained from the buccal surface of these teeth crowns (Appendix A). The enamel surface of the resulting blocks was ground flat with silicon carbide papers (320, 600, and 1200 Buehler, Lake Bluff, IL, USA) under copious water-refrigeration and polished with wet felt paper using diamond spray (0.75 µm; Buehler). Such a procedure was controlled with a caliper until the outer layer of enamel surface was reduced approximately by 100 µm. After being polished, five microhardness indentations (Knoop diamond, 25 g, 10 s, HMV-2000; Shimadzu Corporation, Tokyo, Japan), at distances of 100 µm from each other, were performed in the center of enamel surfaces to ensure specimens were equally allocated into two groups (n = 120/groups: i.e., intact and demineralized enamel), with baseline surface hardness (SH) ranging from 340 to 390 Knoop Hardness (KHN) [32]. Forty samples (*n* = 20/group) were randomly chosen for the baseline gravimetrical analysis and after being let dry for 48 h, they had their baseline dry mass (M) determined in analytical balance (ML203T, Mettler Toledo, Columbus, OH, USA). Half of the teeth fragments (i.e., *n* = 100 for hardness and *n* = 20 for gravimetry) remained intact to receive further treatment; while the other half (i.e., *n* = 100 for hardness and *n* = 20 for gravimetry) were submitted to an artificial demineralizing protocol that aimed to induce a chemical caries lesion [32] (Figure 3).

#### 4.2.2. Enamel Blocks Chemical Demineralizing-Challenge Protocol

One-hundred and twenty (*n* = 120) enamel blocks were submitted to a protocol that was shown to develop caries-like artificial lesions on enamel by immersion in 50 mM buffer acetate solution (1.28 mmol/L Ca(NO_3_)_2_·4H_2_O; 0.74 mmol/L NaH_2_PO_4_·2H_2_O; 0.03 ppm F) at pH 5.0, 37 °C, for 16 h [33]. The microhardness of demineralized enamel was taken again. Specimens with a mean around 80–110 KHN were selected and allocated for the experimental groups (Figure 3). Samples previously assigned for gravimetrical analysis were let to dry out for 48 h for the assessment of dry mass after the specimens’ surface chemical demineralizing challenge (M lesion). 

#### 4.2.3. Treatment

Intact (mean KHN 369 ± 15) and chemically demineralized enamel specimens (mean KHN 93 ± 7.5) were randomly assigned to be treated with one of the following solutions: 0.01% LC, 0.03% LC, 0.05% NaF or ultrapure water (pH 6.8). Briefly, 5µL aliquots of each solution were pipetted on the enamel surface, kept undisturbed for 5 minutes, and then dried with laboratory tissue-paper (Kleenex®, Kimberly-Clark, Irving, TX, USA). This procedure was repeated twice a day, before and after the immersion of each specimen in the demineralizing solution of pH-cycling protocol (2.0 mmol/L Ca(NO_3_)_2_·4H_2_O; 2.0 mmol/L NaH_2_PO_4_·2H_2_O; 0.075 mmol/L acetate buffer, 0.02 ppm F). After 6h of immersion in the demineralizing solution (pH 4.7 at 37 °C), specimens were immersed for 18 h (pH 7.0 at 37 °C) in a remineralizing solution (1.5 mmol/L Ca(NO_3_)_2_·4H_2_O; 0.9 mmol/L NaH_2_PO_4_·2H_2_O; 150 mmol/L KCl; 0.1mol/L Tris buffer; 0.03 ppm F). These procedures (i.e., treatment with solutions and pH-cycling) were repeated for 6 days [33].

#### 4.2.4. Post-Treatment Microhardness and Gravimetrical Measurements

After treatment, a final hardness test (SHf) was conducted following parameters previously described. The percentage of surface hardness change (%SHC) for the intact and demineralized enamel was independently calculated, as follows: %SHC intact enamel = [(SHf − SH)/SH] × 100; and as percentage of hardness surface recovery (%SHR) for demineralized enamel; %SHR = [(SHf − SH lesion)/(SH − SH lesion)] × 100. All tested specimens were longitudinally sectioned and polished to assess their cross-sectional hardness (CSH) (Appendix A). Likewise, for sample gravimetrical measurements, the percentage of dry mass change (%MC) for intact and chemically demineralized specimens was independently calculated, as follows: %MC intact enamel = [(Mf − M)/M] × 100; and %MC demineralized enamel = [(Mf − M lesion)/(M − M lesion)] × 100.

#### 4.2.5. Statistical Analysis

Assumptions of equality of variances and normal distribution of errors were checked for all variables using Levene’s tests. As those assumptions were satisfied for %SHR, cross-sectional, and %MC, three independent one-way ANOVA were performed for each of these variables followed by Tukey tests. Conversely, a failure in the equality of variance test for %SHC for intact enamel led these data to be analyzed by a Kruskal–Wallis test (IBM—SPSS, NY, USA). All analyses were carried out at α = 0.05.

### 4.3. Experiment 2—In Situ/Ex Vivo, Co-Twin Control Study

#### 4.3.1. Experimental Design and Ethical Approval

This experiment investigated the effect of LC in comparison to NaF on the in situ/ex vivo remineralization of chemically demineralized enamel (Figure 4). A double-blind (for volunteer and examiner regarding type of treatment), randomized experiment was designed based on a co-twin control model. The experiment comprised one in situ period of two consecutive weeks, with one in-between weekend wherein treatments and participants assessment were not performed. Twins wore palatal appliances with 4 demineralized enamel blocks each. All experiments of the in situ study were conducted between March and June 2012. The study protocol was approved by the local IRB Committee (protocol #2146, 2010; State University of Montes Claros—UNIMONTES, MG, Brazil) and conformed with the principles outlined in the Helsinki Declaration. Participants and/or their legal guardians signed off written informed consent forms before inclusion in this study.

#### 4.3.2. Demographics and Caries Risk Assessment of the Study Population

Participants were twins from families of a low socioeconomic level who resided in the urban setting of the city of Montes Claros, State of Minas Gerais, Brazil. The water supplies of the city had fluoride levels of < 0.2 ppm, as assessed at the time this study was conducted. The sample consisted of 29 pairs of twins (*n* = 58), who at baseline were 12–24 years old. Assessment of zygosity revealed 9 pairs of monozygotic (MZ) twins, 20 pairs of dizygotic (DZ) twins, and 1 set of DZ triplets (Appendix A). Participants did not present with cavitated lesions prior to study commencement. Accordingly, they also did not exhibit gingival inflammation that could have affected study results due to significant overgrowth of undisturbed dental biofilms. Diet counseling was not performed before or during the study, but participants were asked to provide a diary with descriptions on food and drink consumption corresponding to main meals and snacks. Each twin pair was its own control within the same household.

#### 4.3.3. Intraoral Appliances Preparation and Assembling

Acrylic intraoral palatal appliances were prepared with bilateral lodges (5 × 5 × 3 mm thick), two on the left and two on the right side, to accommodate enamel blocks. Enamel specimens (4 × 4 × 2.5 mm thick) were obtained from the crown (buccal surface) of bovine teeth, baseline assessed for surface baseline microhardness (SH), followed by image acquisition using a quantitative induced-fluorescence (QLF) system [38]. The enamel surfaces of samples were subjected to an artificial demineralizing chemical challenge protocol and assessed for demineralized surface microhardness (SH lesion), as previously described. Demineralized blocks were carefully and randomly fixed with wax onto the lodges of intraoral appliances, ensuring that only the enamel surface of these blocks was exposed (Figure 4). Mounted appliances were sterilized by exposure to ethylene oxide gas. 

#### 4.3.4. In Situ Phase

Three mouthwash solutions were prepared, namely 0.03% LC, 0.05% NaF, and placebo (ultrapure water) with sterilized ultrapure water with pH adjusted at 6.8 (Appendix A). For the first 24 h, all participants were asked to wear the intraoral appliance to allow formation of a salivary acquired pellicle over enamel blocks [5]. A twin member of a pair was randomly assigned to treatment with 0.03% LC, while the other twin to a control treatment consisting of rinsing with 0.05% NaF or placebo solution (Figure 4). Participants were instructed to mouthwash with 15 mL of the correspondent rinsing solution (simply identified with participant’s nickname) for 45 seconds, twice a day (all mornings under supervision and at night before bedtime) for 2 weeks, and only during weekdays (5 days/week). Participants were solicited not to exchange appointed rinsing solutions. The intraoral palatal appliances were removed only during meals (with maximum 1 h duration each and interval between meals of 2–3 h). Participants were advised to perform their routine oral hygiene practices after the meals and before reinserting the appliance in the mouth.

#### 4.3.5. Enamel Final Hardness and QLF measurements

After the 2-week in situ phase, enamel fragments were carefully removed from the appliances, and fixed on acrylic supports for assessment of their final surface hardness (SHf) as previously described. The surface hardness change after treatment, indicating the percentage of surface hardness recovery, was determined as [(SHf − SH lesion)/(SH − SH lesion)] × 100 for each block; then, the mean value of the %SHR of 4 blocks for each volunteer was calculated. Then, a second set of QLF images were captured (QLF-t). A customized QLF setup was employed using a system comprising a light box containing a xenon lamp with a blue filter (λ = 488 nm, 10–20 mW/cm^2^) and a handpiece-camera coupled to a computer. Images were captured and analyzed with software provided by the manufacturer (Inspektor™ Pro Software, version 2.0.0.32, Amsterdam, The Netherlands). Once the image of the enamel surface was captured, a quantitative assessment of the demineralization status was determined by the software (Inspektor Research Systems BV), which uses the pixel values of baseline QLF to reconstruct the surface of a given sample and then subtract those pixels that are considered to be the demineralized structure. A video-repositioning tool of the software was used to ensure that baseline and post-treatment images were captured with a correlation factor higher than 0.90. The difference between the green pixel values in the reference area and those in the lesion area was divided by the reference area (green pixel values) at baseline and after treatment. This difference was expressed as a change in fluorescence. Fluorescence loss (∆F), size of the lesion (area in mm^2^), and fluorescence loss integrated over the lesion size [ΔQ = (∆F × area)] were calculated using a 5% threshold by a single, blinded (for treatment), and calibrated examiner. The values of ∆F and ΔQ of 4 blocks for each volunteer were averaged. 

#### 4.3.6. Statistical Analysis

Three twin pairs (total *n* = 6 individuals) did not come back to follow up. Twenty twin pairs (including two twins of the triplets, total *n* = 40 individuals) were assigned for comparisons between LC and NaF solutions that were performed to assess the effects on the mineralization of enamel. A subset of 7 twin pairs (total *n* = 14 individuals) was evaluated for comparisons between LC and placebo solutions. As assumptions of equality of variances and normal distribution were satisfied, ANOVA and Tukey’s tests were performed for each of the response variables (%SHR, ∆F, and ΔQ) (Minitab®17, Minitab Inc, PA, USA). Analyses were carried out at α = 0.05.

## 5. Conclusions

Overall, the results lead us to conclude that the extract of marine algae *Lithothamnion calcareum,* derived from the Atlantic waters of the northeast coast of Brazil, exerted an equivalent to superior modulatory effect on the enamel de-remineralization dynamics when compared to sodium fluoride. *Lithothamnion calcareum* should be further explored as an alternative to manage oral conditions (i.e., dental caries and erosion) involving enamel demineralization.

## 6. Patents

The work described in this manuscript has formed the core of a patent application to the USPTO that was awarded in 2019-05-14, i.e., Composition for Promoting Oral and General Health and Method for Forming and Using the Same, US Patent No. 10,285,929 B1, therefore acknowledging the novelty of the approach for the promotion of oral health, employing naturally occurring compounds originating from the sea.

## Figures and Tables

**Figure 1 marinedrugs-21-00109-f001:**
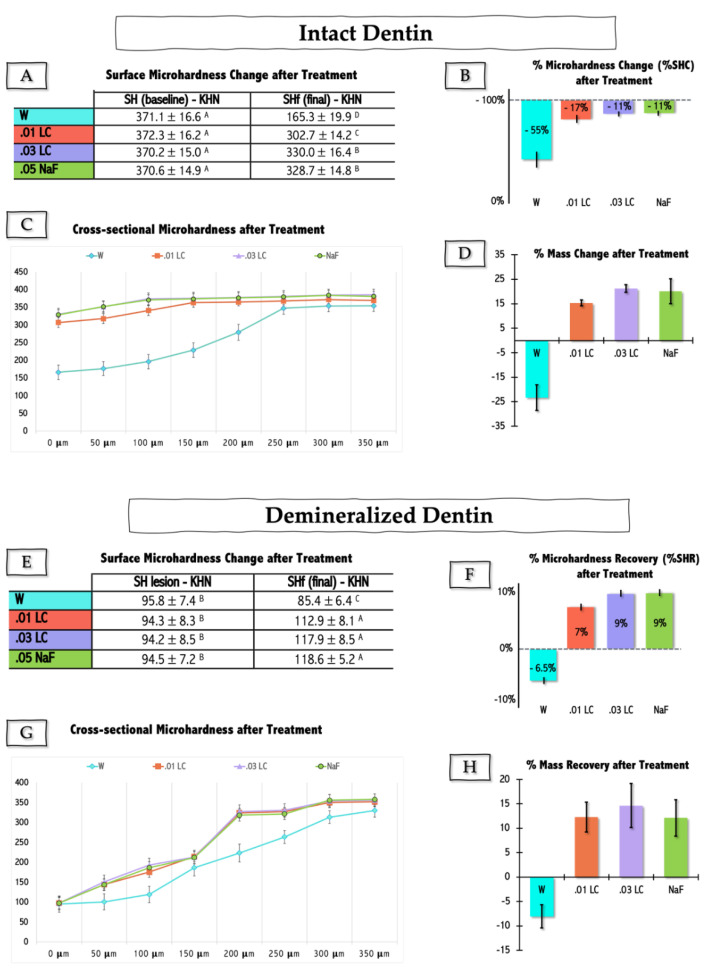
In vitro effect of *Lithothamnion calcareum* (LC) and sodium fluoride (NaF) on enamel de-mineralization dynamics. (**A**,**B**): changes in surface and (**C**) cross-sectional microhardness and mass (**D**) of intact enamel specimens. (**E**,**F**): changes in surface and (**G**) cross-sectional microhardness and mass (**H**) of demineralized enamel specimens. Values are mean and standard deviation (parametric analyses) and mean ± standard error (non-parametric analyses). SH (Baseline) = initial surface hardness of intact enamel; SH lesion = surface hardness of chemically demineralized enamel; SHf = final enamel hardness after treatment. W = treated with water; 0.01 LC = treated with 0.01% LC; 0.03 LC = treated with 0.03% LC; NaF = treated with 0.05% NaF.

**Figure 2 marinedrugs-21-00109-f002:**
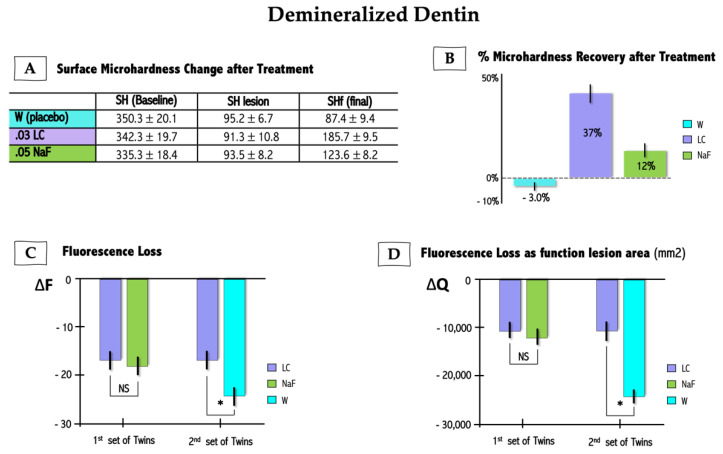
In situ/ex vivo effect of *Lithothamnion calcareum* (LC) and sodium fluoride (NaF) on enamel re-mineralization dynamics. (**A**): change in microhardness surface; (**B**): % microhardness recovery (%SHR) after treatments; and (**C**): fluorescence loss (ΔF) and (**D**) fluorescence loss as function of lesion area (ΔQ) assessed by quantitative induced-fluorescence (QLF). Values are mean and standard deviation (parametric analyses) and mean and standard error (non-parametric analyses). NS = no statistical significance; * = statistical significance. SH (Baseline) = baseline surface hardness of enamel before chemical demineralization; SH lesion = surface hardness of enamel after chemical demineralization; SHf = final surface hardness of demineralized enamel after treatment. W = treated with water; 0.03 LC = treated with 0.03% LC; NaF = treated with 0.05% NaF.

**Figure 3 marinedrugs-21-00109-f003:**
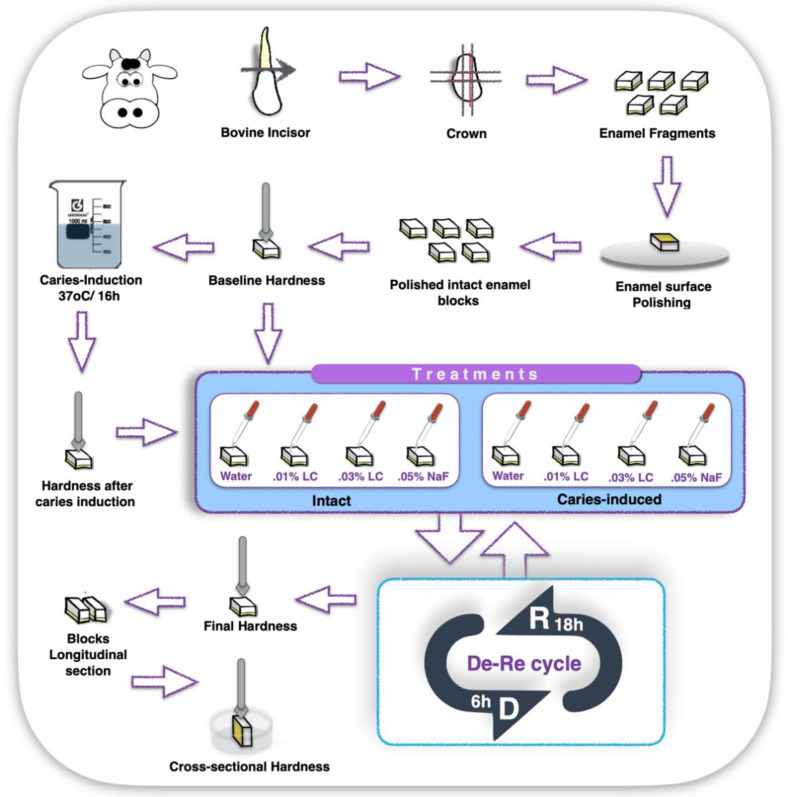
Illustrative schematic of experiment 1—In vitro study. Sequence shows the sectioning of bovine tooth; obtained enamel fragments; enamel surface polishing; enamel surface baseline hardness measurement; chemical induction of caries lesion (demineralization) of half of total sample, while the other half remained intact; surface hardness of demineralized samples; treatment of intact and demineralized samples with Water; 0.01%LC, 0.03% LC or 0.05% NaF; samples demineralization–remineralization cycle (De–Re cycle) for 6 days; enamel surface final hardness; longitudinal sectioning of samples for cross-sectional hardness measurements; and samples cross-sectional hardness measurements.

**Figure 4 marinedrugs-21-00109-f004:**
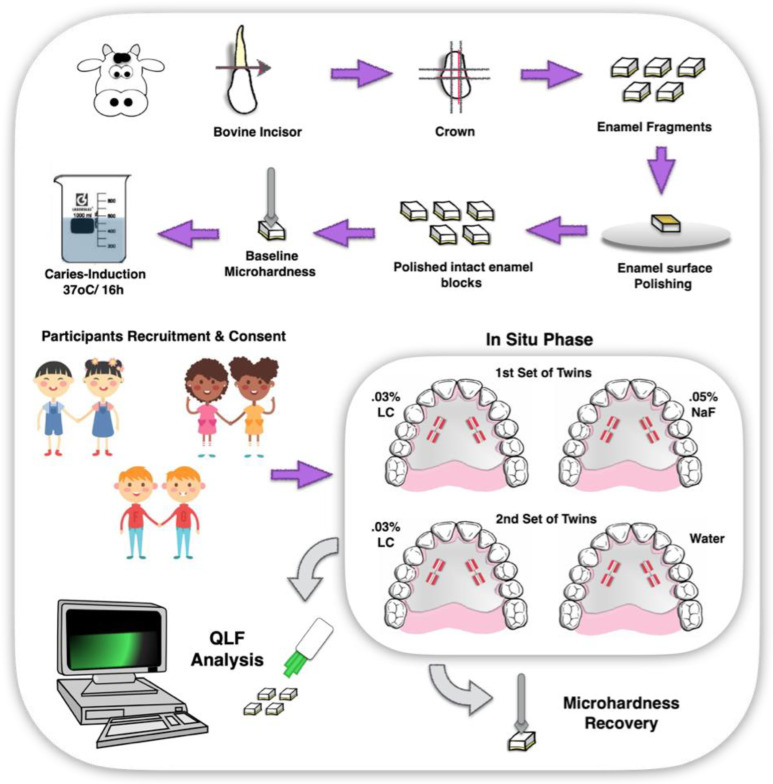
Illustrative schematic of experiment 2—in situ/ex vivo co-twin control study. Sequence shows the sectioning of bovine tooth; obtained enamel fragments; enamel surface polishing; enamel surface baseline hardness measurement; enamel chemical lesion induction (demineralization); participants recruitment and consent; preparation of intraoral appliances for in situ phase; in situ phase in two sets of twins for comparsion of treaments 0.03% LC × 0.05% NaF and 0.03% LC × water; removal of enamel blocks from intraoral appliances after in situ phase; QLF measurements.

## Data Availability

All data generated or analyzed during this study are included in this article and its Appendix A. Further enquiries can be directed to the corresponding author.

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
