# Peer review of "Red Marine Algae Lithothamnion calcareum Supports Dental Enamel Mineralization"

_marinedrugs, 2023, doi:10.3390/md21020109_

Round 1

Reviewer 1 Report

The present paper has a large biotech potential in odontology. The data strongly suggest that the multi-oligoelemental composition of the red marine algae Lithothamnion calcareum (LC) may act in synergism with Ca and Mg to support enamel remineralization.The present results pro­vide compelling evidence that a multi-mineral natural approach can be beneficial to target the dental hard tissue disorders.  LC is not only a safe and bioactive agent for management of dental caries and erosion but can be also envisioned as more inclusive oral healthcare promotor.

Author Response

We would like to thank this reviewer for the kind words of encouragement. We really appreciate such support.

We have reviewed the manuscript for typos and language adjustment. Changes made in the revised version of our manuscript are highlighted in red. 

Reviewer 2 Report

It would be interesting if Red Marine Algae Lithothamnion calcareum were explored as an a formula Instead of  flu-9 oride-based formulations  to improve dental health.

1. This manuscript use the tructure of "introduction - Result - discussion - method - conclusion" has beyond comprehension. Is it possible to follow the conventional form of "introduction - method - structure - discussion - conclusion"?

2.Figure 1 and Figure 2 image formats need to be corrected. In fact, the graph1-A and graph 2-A in the article should be processed and analyzed in tabular form.

3.It is recommended that the preamble be refined to highlight the focus of the research and skip too much about the introduction to tooth enamel demineralization and the controversy over the use of fluoride. At the same time, the concluding paragraph of the preamble can briefly discuss the significance of this study.

4.In subsection 4.3.2, the influence of oral hygiene habits was not considered in the control factors of volunteers.

5.In subsection 4.3.3, generally, the probability of demineralization is different for different tooth positions, so please specify whether the placement of the appliance will have an effect on the change of tooth enamel hardness.

6. The work described in this manuscript has formed the core of a patent application to the USPTO that was awarded in 2019, US Patent No. 10,285,929.    The author have not provide the patent and I can't download the patent either. I cannot confirm the relationship between this patent and this study. The editor needs to judge whether it is appropriate to publish an article after the patent has been granted.

Author Response

We appreciate the positive critiques made by this reviewer.

We made pertinent modifications in the manuscript (highlighted in red) to follow up most of this reviewer's comments/suggestions and provided the following responses to justify points with which we respectfully disagree and/or trust should not be changed to firmly address this study primary objectives.

 Suggestions for Authors

It would be interesting if Red Marine Algae Lithothamnion calcareum were explored as an a formula Instead of  flu-9 oride-based formulations  to improve dental health. Our response: Although we did not quite understand what exactly this reviewer is suggesting here, we want to assure him/her that we have not modified whatsoever the Lithothamnion calcareum composition. Instead, we use the skeletal remains of red marine algae Lithothamnion calcareum as supplied. Thus, we trust, we have truly tested the effect of this marine natural extract on enamel de-remineralization process.

  1. This manuscript use the tructure of "introduction - Result - discussion - method - conclusion" has beyond comprehension. Is it possible to follow the conventional form of "introduction - method - structure - discussion - conclusion"? Our response: We understand this may be an unusual structure for a manuscript. However, we have simply followed the template provided by Marine Drugs, which recommends such a structural sequence of: Introduction, Results, Discussion, Material and Methods and Conclusions. 
  2. Figure 1 and Figure 2 image formats need to be corrected. In fact, the graph1-A and graph 2-A, in the article should be processed and analyzed in tabular form. Our response: Again, though we did not quite understand what exactly this reviewer asked here, we do agree that Figure 1 and Figure 2 should be enhanced. We essentially modified the colors that identify each of the experimental groups. We trust these edited figures are now more accurately contrasting differences and/or showing similarities among these groups.
  3. It is recommended that the preamble be refined to highlight the focus of the research and skip too much about the introduction to tooth enamel demineralization and the controversy over the use of fluoride. At the same time, the concluding paragraph of the preamble can briefly discuss the significance of this study.  Our response: We believe that the Marine Drugs readership may not be necessarily aware on current therapies and potential controversies relative to the management of dental caries and dental erosion. Thus, this study introduction was designed to provide Marine Drugs readers with an overall background on this topic and finally justify why it is important to procure natural alternative to fluorides. Therefore, we retain that the major idea of this introduction should be maintained. Anyway, we have  slightly modified this section to address the significance of this study.
  4. In subsection 4.3.2, the influence of oral hygiene habits was not considered in the control factors of volunteers. Our response: We have added a paragraph in Discussion to reflect this reviewer's concern.
  5. In subsection 4.3.3, generally, the probability of demineralization is different for different tooth positions, so please specify whether the placement of the appliance will have an effect on the change of tooth enamel hardness. Our response: We are very sorry, but we could not understand what this reviewer is requiring here. Pre-demineralized enamel blocks had been always mounted to have their buccal surface exposed on the surface of oral appliances. There was no variation whatsoever of this enamel block placement pattern among the appliances.
  6. The work described in this manuscript has formed the core of a patent application to the USPTO that was awarded in 2019, US Patent No. 10,285,929.    The author have not provide the patent and I can't download the patent eitherI cannot confirm the relationship between this patent and this study. The editor needs to judge whether it is appropriate to publish an article after the patent has been granted. Our response: We have added exact  date of publication and missing alphanumeric code for this patent. In addition, we attached a copy of it in here to facilitate this reviewer and editor assessment (please see attachment). 

Reviewer 3 Report

1

1)      L75-76: The abbreviations, %SHC and %SHR are not clear.

2)      Figure 1: The Y and X-axis heading are missing.

3)      Figure 1: The alphabetical labelling of the sup-figures seems missing.

4)      Figure 1 &2: Make sure all the plots indicate with both lower and upper whiskers.  

5)      The full term for KHN must be provided.

6)      L229: Missing full stop.

7)      Figure 3 & 4: The process must be clearly explained in the figure legend.

8)      Figure 3: ‘Des-Re’ seems got typo.

9)      L309: Remove ‘ethical approval’ from the heading since mentioned in L413.

10)   Is this study registered under clinical trial prior to testing in human?

Author Response

We appreciate reviewer comments and suggestions. I trust that suggested modifications improved our manuscript. All changes made in the manuscript are shown in red font.

Here our point-by-point answer to this reviewer comments and concerns:

1)      L75-76: The abbreviations, %SHC and %SHR are not clear.

Our response: We reviewed the text to make sure all abbreviations are clear and addressed what they meant to indicate.

2)      Figure 1: The Y and X-axis heading are missing.

Our response: We guess that reviewer is referring to the figure in the panel that represents results of % Mass Change after Treatment for intact and artificially demineralized enamel. We have completely modified the layout of this graph which now appears with x and y axes. 

3)      Figure 1: The alphabetical labelling of the sup-figures seems missing. 

Our response: Very good observation. We have added alphabetical labelling of the sup-figures.

4)      Figure 1 &2: Make sure all the plots indicate with both lower and upper whiskers.  

Our response: While we agree that deviation bars should be included in all graphs they are applicable, we do not necessarily think that lower (for negative module values) or upper (for positive module values) "whiskers" should be included. Anyways, to satisfy this reviewer inquiry, we have modified figures accordingly.

5)      The full term for KHN must be provided.

Our response: We have amended it. 

6)      L229: Missing full stop.

Our response: Thanks for noticing it. We have included the full stop. This statement is now in L239.

7)      Figure 3 & 4: The process must be clearly explained in the figure legend.

Our response: We have modified legend of figure 3 and 4 as suggested.

8)      Figure 3: ‘Des-Re’ seems got typo.

Our response: We have amended typo which is now read as De-Re.

9)      L309: Remove ‘ethical approval’ from the heading since mentioned in L413.

Our response: We have not removed term as the first time it appears is in the heading of sub-section. Moreover, other reviewers asked us to highlight description of the protocol ethical approval. 

10)   Is this study registered under clinical trial prior to testing in human?

Our response: No, this study has not been registered under a clinical trial repository. We want to offer this reviewer a justification for it. The protocol and experiments described in this study were approved and conducted almost 12 years ago. Results are only now disclosed because we have decided to first apply for the patent concerning this material. As this reviewer is certainly aware, the registration for clinical trials has been only more recently required. We acknowledge the potential benefits of registering clinical protocols before the initiation of data collection, which unfortunately we could not accomplish in this case for the reason above mentioned.

Reviewer 4 Report

Dear authors,

thank you for your well structured study. Please address the following remarks:

1.       Consider to move M&M section between introduction and results

2.       Consider to structure your abstract in accordance to the M&M and results section

3.       Please post results (with p-values) in the abstract

4.       Please add information upon the polishing procedure to the M&M section. Regret from pointing towards supplementary files!

5.       Please post identification number and date of the ethical approval

6.       Please provide address and name of the ethical committee

7.       Apply equal headings for “intact dentin” and “demineralized dentin” in figure 1

8.       Apply equal heading for “demineralized enamel” in figure 2.

Author Response

We appreciate reviewer comments and suggestions. I trust that suggested modifications improved our manuscript. All changes made in the manuscript are shown in red font.

Here our point-by-point answer to this reviewer comments and concerns:

Dear authors,

Thank you for your well structured study. Please address the following remarks:

1. Consider to move M&M section between introduction and results

Our response: We absolutely agree with this reviewer that presenting M&M before results and discussion sounds reasonable. However, we just followed the Marine Drugs manuscript template, wherein M&M is presented after Discussion and before Conclusion. We wish we could consider this reviewer suggestion, but it seems we do not have an option regarding this format.

2. Consider to structure your abstract in accordance to the M&M and results section

Our response: We have re-read the Abstract and regret saying we do not quite understand what this reviewer meant with that we should structuring it in accordance to the M&M and results. We made modifications to the Abstract and hope this looks in accordance to the M&M and results section.

3. Please post results (with p-values) in the abstract

Our response: We understand this reviewer point, but respectfully refrain to include the p-values in the abstract. As the reviewer knows we performed different experiments and ran statistical analysis for each experiment. This means we have different correspondent p-values. Rather than presenting a tedious description reflecting every experiment and outcomes, our approach for the abstract was to provide readers with the big picture of the study as far objectives, results and conclusions. We thank and hope this reviewer understand our choice.

4. Please add information upon the polishing procedure to the M&M section. Regret from pointing towards supplementary files!

Our response: We have amended it. 

5. Please post identification number and date of the ethical approval

Our response: We have included IRB protocol number and year.

6. Please provide address and name of the ethical committee

Our response: We have included the name of the ethical committee in the M&M (State University of Montes Claros - UNIMONTES, Montes Claros, MG, Brazil). The IRB physical address is: Campus Universitário Professor Darcy Ribeiro – Avenida Rui Braga, S/Nº – Vila Mauricéia Montes Claros CEP 39401-089, MG, Brazil. We refrain to add the address in the paper as it does appear to be a routine information included in manuscripts.

7. Apply equal headings for “intact dentin” and “demineralized dentin” in figure 1

Our response: We have amended it. 

8. Apply equal heading for “demineralized enamel” in figure 2.

Our response: We have amended it. 

Round 2

Reviewer 4 Report

Dear authors,

thank you for your corrections. There is only one minor request. Please change the front styles of the headings in figure 1 and 2 (“intact dentin, demineralized dentin”). Please use the same style as in the main text body (Palatino Linotype?).

Author Response

We thank this reviewer for further assessment of our manuscript.

We have followed requested changes and enclosed updated manuscript along with Figure 1 and 2.
